# Urban scaling of opioid analgesic sales in the United States

Pricila H. Mullachery [1]* , Usama Bilal [1,2]

**1** Urban Health Collaborative, Drexel Dornsife School of Public Health, Philadelphia, Pennsylvania, United States of America, **2** Department of Epidemiology and Biostatistics, Drexel Dornsife School of Public Health, Philadelphia, Pennsylvania, United States of America

☯ These authors contributed equally to this work.
* phm32@drexel.edu

## Abstract

Opioid misuse is a public health crisis in the United States. The origin of this crisis is associated with a sharp increase in opioid analgesic prescribing. We used the urban scaling framework to analyze opioid prescribing patterns in US commuting zones (CZs), i.e., groups of counties based on commuting patterns. The urban scaling framework postulates that a set of scaling relations can be used to predict health outcomes and behaviors in cities. We used data from the Drug Enforcement Administration's Automated Reports and Consolidated Ordering System (ARCOS) to calculate counts of oxycodone/hydrocodone pills distributed to 607 CZs in the continental US from 2006 to 2014. We estimated the scaling coefficient of opioid pill counts by regressing log(pills) on log(population) using a piecewise linear spline with a single knot at 82,363. Our results show that CZs with populations below the knot scaled superlinearly ($\beta = 1.36$), i.e., larger CZs had disproportionally larger pill counts compared to smaller CZs. On the other hand, CZs with populations above the knot scaled sublinearly ($\beta = 0.92$), i.e., larger CZs had disproportionally smaller pill counts compared to smaller CZs. This dual scaling pattern was consistent across US census regions. For CZs with population below the knot, the superlinear scaling of pills is consistent with the explanation that an increased number of successful matches between prescribers and users will lead to higher prescribing rates. The non-linear scaling behavior observed could be the result of a combination of factors, including stronger health care systems and prescribing regulation in largely populated commuting zones, as well as high availability of other opioids such as heroin in these commuting zones. Future research should explore potential mechanisms for the non-linearity of prescription opioid pills.

## Introduction

Opioid misuse is a public health crisis in the United States where opioid overdoses have increased four-fold in the past 20 years [1]. The origin of this crisis is associated with a sharp increase in opioid analgesic prescribing, with prescriptions of oxycodone increasing more than five-fold between 1999 and 2011 [2]. Increases in opioid prescribing rates was in large

**Data Availability Statement:** Data and code are available in a public repository: https://github.com/usamabilal/ARCOS_Pill_Scaling/.

**Funding:** This research was supported by Office of the Director of the National Institutes of Health under award number DP5OD026429. This grant

was awarded to UB. The funders had no role in study design, data collection and analysis, decision to publish, or preparation of the manuscript.

**Competing interests:** The authors have declared that no competing interests exist.

part the result of aggressive marketing tactics by the pharmaceutical industry [2] and changes in guidelines for the treatment of chronic pain [3].

Small and rural communities are at the epicenter of the opioid crisis. A journalistic investigation by the Washington Post in 2019 revealed that, between 2006 and 2012, small and rural towns received a disproportionally large number of oxycodone and hydrocodone pills, two of the main analgesic opioids used in the US [4]. However, studies examining patterns across rural and urban counties found that higher density of pharmacies and availability of prescribers, both present in urban areas, are associated with higher prescribing rates [5, 6]. The finding of higher prescribing rates in urban communities is also consistent with the idea that people living in rural areas of the country are more likely to face barriers to access health care [7]. In this context, it is not clear whether urbanicity or population size have a role in the distribution of pills across the country. This paper uses the urban scaling framework to examine the relationship between population size and opioid prescribing patterns in the US.

Scaling is the response of complex systems, such as cities, to variation in their size [8]. The application of this framework has previously shown that a set of scaling relations can be used to predict several features of cities [9]. According to this framework, sublinear scaling is observed when features of cities, e.g. number of gas stations, length of the road network, are disproportionately smaller in larger versus smaller cities, as a consequence of economies of scale [9]. On the other hand, cities also exibit superlinear scaling, i.e., when outcomes such as economic productivity and creative outputs are disproportionally larger in larger versus smaller cities. For example, larger cities have a disproportionately higher economic productivity compared to smaller cities [10]. This phenomenon is the result of the densification of social networks due to an increase in population size; high number of social conections in large cities leads to a disproportional increase in various outcomes such as economic productivity and number of patents [11].

Health outcomes also show scaling behaviors [12–14]. Non-communicable conditions such as diabetes and obesity exhibit sublinear scaling in the US, i.e., relatively less common in larger cities, possibly due to better access to resources and medical services in large cities [12]. On the other hand, sexually transmitted infections (i.e., chlamydia, gonorrhea and syphilis] exhibit superlinear scaling, i.e., relatively more common in larger cities [13, 14]. This type of superlinear behavior may be the result of a disproportionally larger number of social connections in dense urban centers, which in turn increases the changes of successful matches between cases and susceptible individuals [15]. Similar mechanisms related to differences in access to resources and number of social connections between large and small cities may play a role in the distribution of opioid pills across the US. In this paper, we estimate the scaling parameter for opioid analgesic pills distributed across 607 US Commuting Zones (CZs).

## Materials and methods

We used data from the Drug Enforcement Administration's Automated Reports and Consolidated Ordering System (ARCOS), made available by the Washington Post, to calculate counts of oxycodone/hydrocodone pills distributed to 607 Commuting Zones (CZs), which are all CZs in the continental US, from 2006 to 2014. CZs are aggregations of counties based on commuting patterns. We used CZs because they are more likely to account for the complex networks across counties that share interconnected economies, which may be important to the understanding of the macro-level determinants of opioid outcomes. We excluded CZs that include counties in non-contiguous states (Alaska and Hawaii) because they may not be a good representation of the commuting networks that, in the continental US, often cross state lines. Another advantage of using commuting zones as a spatial unit is that they provide a

more complete picture of the country, from rural to highly urbanized areas. These CZs have a perfect overlap with county boundaries which make it straightforward to aggregate the measures from counties to compute CZ-level measures [16].

## Data sources

Data from the Drug Enforcement Administration's Automated Reports and Consolidated Ordering System (ARCOS) was made available by the Washington Post in July of 2019 and updated in February 2020. Access to the data by the Washington Post was gained as a result of a court order. The data set contains data on shipments of oxycodone and hydrocode pills to chain pharmacies, retail pharmacies and practitioners, including amount distributed and location of the pharmacy/office. Data included only oxycodone and hydrocodone pills. Other opioids were excluded because they were shipped in much lower quantities. The data was cleaned to remove shipments that did not go to consumers such as shipments from one distributer to another. We accessed the Washington Post data through the use of the R package ARCOS (https://cran.r-project.org/web/packages/arcos), which also included population counts by year and county. Data and code used in this analysis can be found here: https://github.com/usamabilal/ARCOS_Pill_Scaling/. Commuting Zones were defined using 2010 boundaries [16]. We used population estimates from the US Census Bureau [17].

## Analysis

First, we calculated the counts of oxycodone and hydrocodone in each CZ by aggregating all pills shipped to pharmacies/offices in counties within the CZ from 2006 to 2014. We also calculated the average population in each CZ from 2006 to 2014, We then estimated the scaling coefficient of opioid pill counts by regressing log(pills) on log(population). We used the following model:

$$\ln(Y_i) = \alpha + \beta * \ln(N_i) + \epsilon_i$$

Where $Y_j$ is the log of the number of pills for the $i$-th CZ and $N_i$ is the log of the population in the CZ. $\beta$ is the scaling coefficient: $\beta < 1$ corresponds to sublinear scaling, i.e., pill counts disproportionally higher in smaller CZs, and $\beta > 1$ corresponds to superlinear scaling, i.e., pill counts disproportionally higher in larger CZs. We used the model described above following standard practice in the urban scaling literature and did not adjust for any variables that may be in the pathway between population size and the outcome.

After visually exploring initial results of the log-log plots, we detected a strong non-linear pattern. A plot of the residuals from the linear scaling model against the population size showed an "U" shape curve with the vertex located around a population of 100,000 (S1 Fig). To acknowledge this lack of linearity, we introduced a piecewise linear spline, which is a version of the power-law with a cut-off model described by Clauset et al. [18]. We looked for the knot position that best fit the data by using the segmented package in R, which looks for the knot position that minimizes the log likelihood resulting from the model with the spline. Based on this, we included a linear spline with a knot at a population of 82,363 (representing the 35[th] percentile of CZ population across our sample). We also checked whether the model with a linear spline had a better fit than the model without a spline by comparing the Akaike Information Criteria (AIC) for each model.

To visually depict the relationship between number of pills and population, we created a plot showing the log of population on the x axis and the log of pill counts on the y axis. To this plot, we added a linear fit with a linear spline at a population of 82,363. We also mapped the residuals ($\epsilon_i$) from the model to the 607 CZs [19]. To explore potential place-specific effects,

we adjusted the models for region where the CZ is located, i.e., Northeast, Midwest, South, and West, by including dummy variables for the regions. We also presented the scaling coefficients for each region individually; we estimated four separate models, one for each region.

## Sensitivity analysis

To test whether our choice of spatial unit had an influence on the scaling behavior, we replicated the analysis using a different spatial definition–the Core-based Statistical Areas (CBSAs), stratified into Metropolitan areas (urban core with population of 50,000 or more) and Micropolitan areas (urban core with population between 10,000 and 50,000 people), using 2013 boundaries. We analyzed data from all 909 CBSAs (377 metro and 532 micro areas). Moreover, during the visual exploration of residuals to ascertain the non-linearity of scaling, we found three strong negative outliers. We repeated the scaling analysis of CZs by (a) excluding those outliers, but keeping the same spline knot (82,363), and (b) excluding those outliers and re-calculating the optimal spline knot (in this case, 151,631, representing the 49.5th percentile of CZ population).

All analyses were conducted in R v4.0.0.

## Results

The scaling coefficient for opioid analgesic pills in all 607 US Commuting Zones from 2006 to 2014 was 1.08 (95% CI 1.05–1.11), corresponding to superlinear scaling. These results show that the number of analgesic opioid pills was disproportionately higher in large (vs. small) CZs (S2 Fig). Specifically, a CZ with 1% larger population had 1.08% greater pill count. However, we found that the model introducing a spline had a better fit than the model without a spline (AIC = 127.8 in the model with a spline vs AIC = 191.3 in the model without a spline), indicating a non-linear scaling behavior. Fig 1 shows that CZs with population below the knot (population of 82,363) scale superlinearly ($\beta = 1.36$, 95%CI 1.23 to 1.50), and CZs with population above the knot scale sublinearly ($\beta = 0.92$, 95%CI 0.88 to 0.95). This means that for CZs below the knot, a 1% larger CZ had a 1.36% higher pill count, while for CZs above the knot, a 1% larger CZ had a 0.92% higher pill count.

Table 1 shows a comparison of the coefficients after adjustment and stratification for census region. These results show that unadjusted and adjusted models had similar coefficients, and that the overall pattern across regions was similar: superlinearity (or weaker sublinearity) in CZs below the knot and sublinearity (or weaker superlinearity) in CZs above the knot. However, CZs in the Midwest Region had a tendency towards a superlinear behavior, with those below the knot showing strong superlinearity ($\beta = 1.42$, 95%CI 1.20 to 1.64) and those above the knot showing linearity ($\beta = 1.00$, 95%CI 0.93 to 1.06). On the other hand, CZs in the Northeast Region showed a tendency towards a sublinear behavior, with those below the knot showing linearity ($\beta = 1.05$, 95%CI 0.81 to 1.30) and those above the knot showing sublinearity ($\beta = 0.94$, 95%CI 0.88 to 1.00).

We also created models with CZs excluding the three negative outliers seen in Fig 1 and models adjusted for percentage of the population 0–14 and 65 and older. The results from these models are seen in S1 Table. They show that our findings were robust to all adjustments, in that patterns were qualitatively similar in direction and significance.

Fig 2 maps the residuals from the model to the 607 CZs in the continental US. The map shows clusters of CZs with higher-than-expected distribution counts in the Appalachians, Ozarks, and Northern California/Southern Oregon.

The scaling parameters using CBSAs, instead of CZs, showed a similar pattern compared to that seen in the main analysis: superlinear scaling for micropolitan CBSAs (core county with

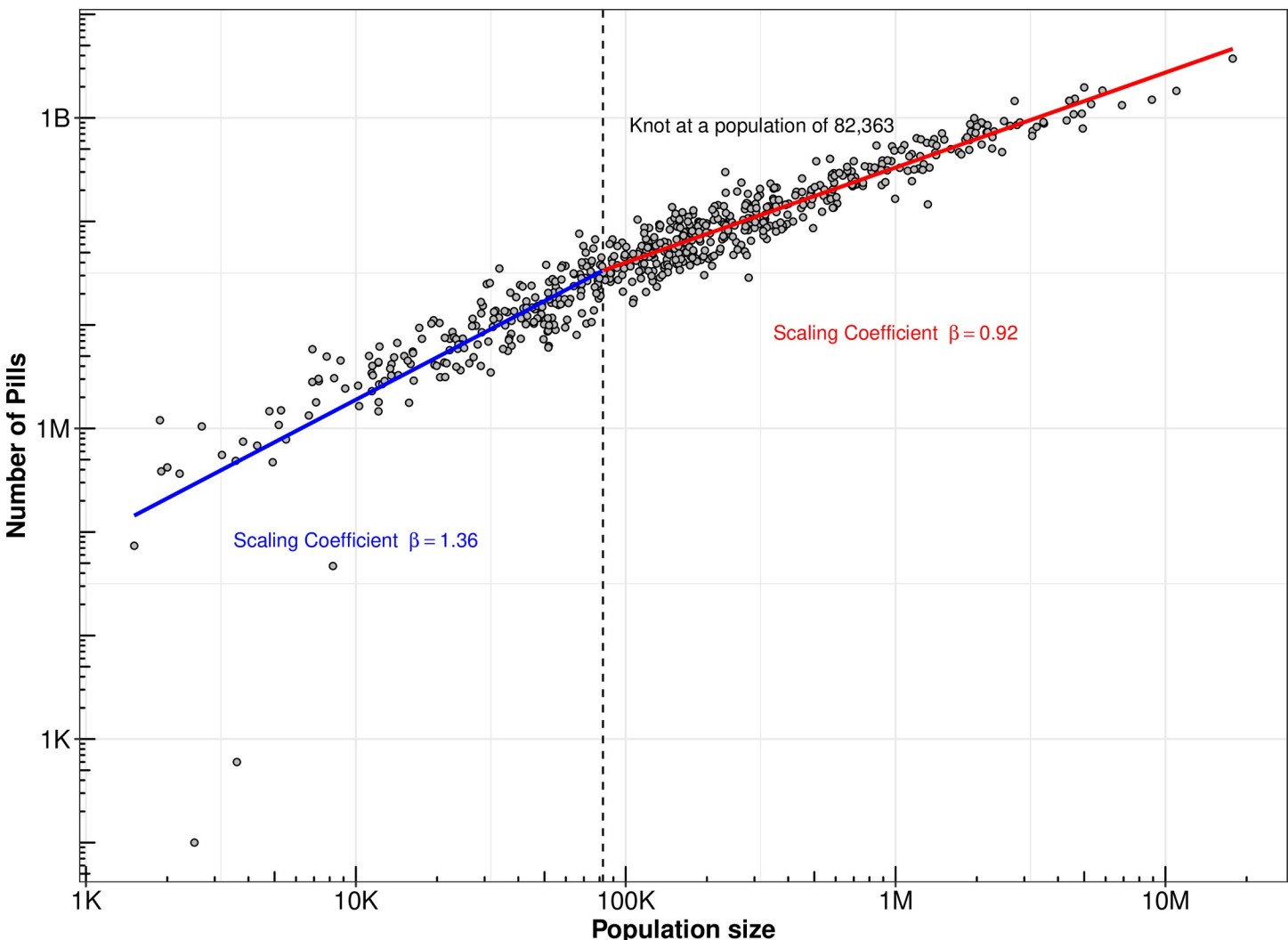

**Fig 1. Non−linear scaling of pill sales including all 607 CZs: Piecewise regression with a spline at a population of 82,363.** Footnote: β is the coefficient of the regression log(pills) on log(population). Sources: ARCOS and Census Bureau.

**Table 1. Scaling coefficients from a piecewise linear model.**

|  | n | β₁ (95% CI)ᵃ | β₂ (95% CI)ᵃ |
|---|---|---|---|
| Unadjusted | 607 | 1.36 (1.23–1.50) | 0.92 (0.88–0.95) |
| Adjusted for Regionᵇ | 607 | 1.35 (1.22–1.49) | 0.92 (0.89–0.96) |
| Stratified by Regionᶜ |  |  |  |
| Midwest Region | 202 | 1.42 (1.20–1.64) | 1.00 (0.93–1.06) |
| Northeast Region | 38 | 1.05 (0.81–1.30) | 0.94 (0.88–1.00) |
| South Region | 248 | 1.17 (1.01–1.33) | 0.87 (0.82–0.92) |
| West Region | 119 | 1.39 (1.10–1.68) | 0.94 (0.88–1.00) |

ᵃ$\beta_1$ and $\beta_2$ are the scaling coefficients below and above the knot (population of 82,363).

ᵇ Models adjusted for region included dummy variables for each region.

ᶜ Stratified models included the CZs for each region separately.

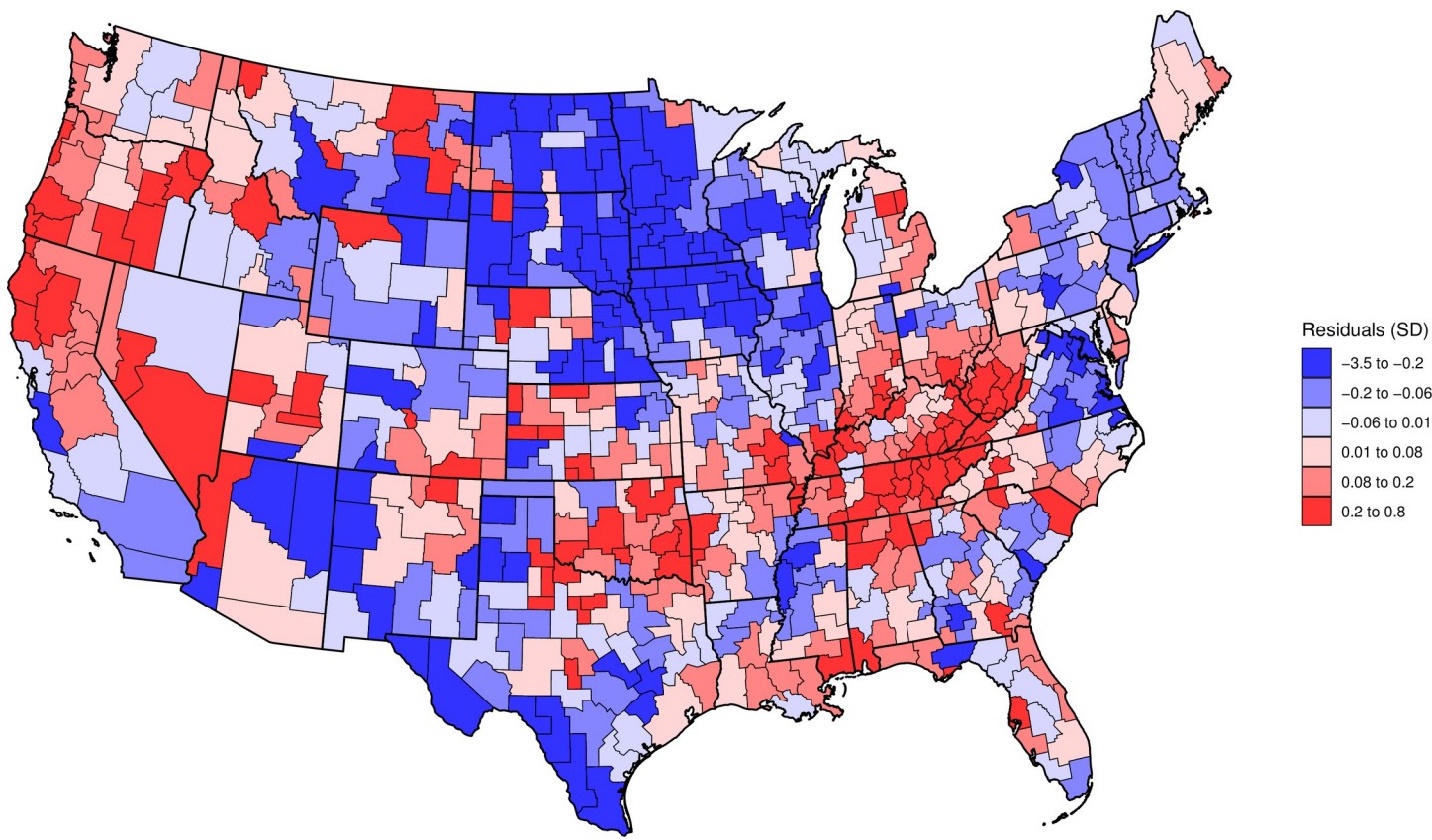

**Fig 2. Map of residuals from the regression log(pills) on log(population).** Footnote: Commuting Zones were defined using 2010 boundaries. Source: ARCOS and Census Bureau.

population between 10,000 and less than 50,000 people) and sublinear scaling for metropolitan CBSAs (core county with population 50,000 or more) (S3 Fig).

## Discussion

In this study on the scaling properties of opioid analgesic sales, we found three key results. First, we found that CZ stratified into two groups, i.e., those below and those above the a population of 82,363 (around the 35th percentile of population), exhibit different scaling behaviors. In CZs with a population below the knot, opioid analgesic pills scaled superlinearly, while in CZs above the knot opioid analgesic pills scaled sublinearly. Second, while we found a consistency in this pattern across census regions, CZs in the Midwest and Northeast Regions displayed a tendency towards more superlinear and sublinear behaviors, respectively. Lastly, we found a spatial distribution of residuals with high sales concentrated in the Appalachians, Ozarks and the West Coast.

Our results show the nonlinear scaling behavior of opioid pills, a pattern that has been found in other outcomes [20]. This nonlinear scaling is driven by a disproportionally large number of pills in mid-sized CZs, which splits the CZs in two groups that can be better represented by generating two scaling coefficients. For CZs with population below the knot, the superlinear scaling of pills is consistent with the explanation that a disproportionally large number of social connections in relatively larger CZs creates an environment that facilitates a disproportionally larger number of successful matches between prescribers and patients,

which in turn may lead to superlinear scaling of opioid analgesic pills [15]. However, for CZs with population at or above the knot, the pattern was inverted. One potential explanation is that the rate of successful matches decreases beyond a certain threshold. This explanation is plausible, but it does not necessarily account for the shift from superlinear to sublinear. Alternatively, the pattern observed could be the result of a combination of factors. Mid-sized CZs may have lower capacity to regulate prescribers and pharmacies, and train health care providers around safe prescription guidelines in comparison with large CZs [21, 22]. In addition, the existence of other opioids such as heroin, which are less expensive than prescription opioids in the underground market, are more likely to be available in larger cities [23]. This could have also contributed to the sublinear scaling of opioid pills in CZs above the knot. Future studies should explore these potential mechanisms with data on prescribers and practitioners and the presence or absence of specific regulations.

Results stratified by geographic region are consistent with the general pattern despite some variations. First, in the Midwest, the strong superlinear pattern in CZs below the knot paired with the linear pattern in CZs above the knot indicates a threshold effect, after which CZs have opioid pill counts that are proportional to their population. Second, the overall sublinear pattern in the Northeast indicates that smaller CZs in this region have a disproportionally high count of opioid pills compared to larger cities. Potential explanations include differences in the profile of opioids used, i.e., prescription opioid vs. heroin and illicitly manufactured synthetic opioids, across these regions [23]. The mapping of results also showed several clusters of high counts in the Appalachians and Ozarks, consistent with other studies [6, 19] that point to high rates of opioid misuse in Southern states in the late 2000's.

Our analysis has limitations. First, studies have shown that the scaling behavior of cities is sensitive to the definition of the spatial unit [24, 25]. This phenomenon, also known as Modifiable Arial Unit Problem (MAUP), is not necessarily regarded as a failure of the urban scaling approach but an expression of the different nature of urban spaces such as city cores, which represent a denser environment, and a larger metropolitan area [25]. We chose commuting zones as our spatial unit because they are more likely to account for the complex networks across counties that share interconnected economies, while also including counties that are not part of a metropolitan area, many of which have elevated number of pills sales. To test whether our choice of spatial unit had an influence on the scaling behavior of opioid sales, we replicated the analysis using a different spatial definition–the Core-based Statistical Areas (CBSA). This analysis showed consistent result with those obtained from commuting zones. Finally, the ARCOS data set contains information on the number of pills distributed to providers and pharmacies and thus we cannot account for patients who may have obtained their medication from prescribers or pharmacies outside of their commuting zones. The fact that we used commuting zones rather than county may have minimized this issue as CZs include a larger area where people live and work.

## Conclusions

Our results point to the potential role of population size in the distribution of opioid analgesic opioid pills. The study period, between 2006 and 2014, was also marked by an increase in the number of drug overdose deaths involving prescription opioids. Thus, future work should examine the links between population dynamics, opioid sales and drug overdose deaths, including testing of potential mechanisms leading to superlinear/sublinear scaling. And as more recent data on opioid analgesic distribution emerge, trends over time must also be examined. Understanding the patterns that emerge from population dynamics may have the potential to inform policies addressing the opioid epidemic. This is even more relevant as illicitly

manufactured synthetic opioids such as fentanyl have been increasingly used in the production of counterfeit opioid analgesic pills [26], where fentanyl powder and pill presses are used to produce pills that resemble oxycodone and hydrocodone pills [23]. Fentanyl-related overdose deaths have spiked since the introduction of illicitly manufactured synthetic opioids in the US drug market around 2013 [26]. Understanding the dynamics of opioid pills distribution in highly urbanized areas may help predict future scenarios in these areas where a large number of people are potentially exposed to opioid misuse.

## Supporting information

**S1 Fig. Residuals from linear scaling model of oxycodone/hydrocodone pills. Source: ARCOS and Census Bureau**[*]. [*]Copyright protection is not available for any work of the United States Government (Title 17 U.S.C., Section 105). Thus, you are free to reproduce census materials as you see fit. We would ask, however, that you cite the Census Bureau as the source. https://www2.census.gov/geo/pdfs/maps-data/data/tiger/tgrshp2019/TGRSHP2019_TechDoc.pdf. Footnote: blue line is a loess smoother of standardized residuals on log(population) including all commuting zones; the red line excludes the three strong outliers.
(TIF)

**S2 Fig. Number of oxycodone/hydrocodone pills distributed across U S Commuting Zones from 2006 to 2014 by population size.** Footnote: β is the coefficient of the regression log (pills) on log(population). Red-colored CZs represent positive residuals and green-colored CZs represent negative residuals. Source: ARCOS (through the Washington Post) and Census Bureau.
(TIF)

**S3 Fig. Non-linear scaling of pill sales stratified by type of Core-based Statistical Areas (CBSA).** Footnote: β is the coefficient of the regression log(pills) on log(population). Micropolitan CBSAs (red) are those built around an urban cluster with population between 10,000 and less than 50,000 people. Metropolitan CBSAs (blue) are those built around urban clusters of 50,000 people or more. Sources: ARCOS and Census Bureau.
(TIF)

**S1 Table. Scaling coefficients from adjusted models compared to unadjusted models.**
(DOCX)

## Author Contributions

**Conceptualization:** Pricila H. Mullachery, Usama Bilal.

**Formal analysis:** Pricila H. Mullachery, Usama Bilal.

**Funding acquisition:** Usama Bilal.

**Writing – original draft:** Pricila H. Mullachery.

**Writing – review & editing:** Pricila H. Mullachery, Usama Bilal.

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
