## [Decision Letter · Decision Letter 0]

19 Oct 2020

PONE-D-20-08832

Urban scaling of opioid analgesic sales in the United States

PLOS ONE

Dear Dr. Mullachery,

Thank you for submitting your manuscript to PLOS ONE. After careful consideration, we feel that it has merit but does not fully meet PLOS ONE’s publication criteria as it currently stands. Therefore, we invite you to submit a revised version of the manuscript that addresses the points raised during the review process.

It is an important topic, and the results would deserve to have a nice publication.

However, the paper still misses a strong development of the state of the art and the reflexion (ofllowing also reviewer 1)

We look forward to receiving your revised manuscript.

Kind regards,

Celine Rozenblat

Academic Editor

PLOS ONE

Journal Requirements:

2.Our internal editors have looked over your manuscript and determined that it is within the scope of our Cities as Complex Systems Call for Papers. This collection of papers is headed by a team of Guest Editors for PLOS ONE: Marta Gonzalez (University of California, Berkeley) and Diego Rybski (Potsdam Institute for Climate Impact Research).

The Collection will encompass a diverse and interdisciplinary set of research articles applying the principles of complex systems and networks to problems in urban science.  Additional information can be found on our announcement page: https://collections.plos.org/s/cities.

If you would like your manuscript to be considered for this collection, please let us know in your cover letter and we will ensure that your paper is treated as if you were responding to this call. If you would prefer to remove your manuscript from collection consideration, please specify this in the cover letter.

4.We note that [Appendix 1 and Figure 2] in your submission contain [map/satellite] images which may be copyrighted. All PLOS content is published under the Creative Commons Attribution License (CC BY 4.0), which means that the manuscript, images, and Supporting Information files will be freely available online, and any third party is permitted to access, download, copy, distribute, and use these materials in any way, even commercially, with proper attribution. For these reasons, we cannot publish previously copyrighted maps or satellite images created using proprietary data, such as Google software (Google Maps, Street View, and Earth). For more information, see our copyright guidelines: http://journals.plos.org/plosone/s/licenses-and-copyright.

1.    You may seek permission from the original copyright holder of Appendix 1 and Figure 2 to publish the content specifically under the CC BY 4.0 license. 

Additional Editor Comments (if provided):

It is an important topic, and the results would deserve to have a nice publication.

However, the paper misses in my point of view the theoretical hypotheses that would strengthen the interpretation. In particular, it does not contextualize the relation between medical treatments / deseases / medical infrastructure. It is argued that there are no studies on scaling and health that is not true: see for example L. E. C. Rocha, A. E. Thorson, R. Lambiotte (2015). The non-linear health consequences of living in larger cities, Journal of Urban Health 92 (5) and some others since that publication....

so please highly strength the state of the art and the reflexion (fllowing also reviewer 1)

Reviewers' comments:

Reviewer's Responses to Questions

**Comments to the Author**

1. Is the manuscript technically sound, and do the data support the conclusions?

Reviewer #1: Yes

Reviewer #2: Yes

2. Has the statistical analysis been performed appropriately and rigorously? 

Reviewer #1: Yes

Reviewer #2: I Don't Know

3. Have the authors made all data underlying the findings in their manuscript fully available?

Reviewer #1: No

Reviewer #2: Yes

4. Is the manuscript presented in an intelligible fashion and written in standard English?

Reviewer #1: Yes

Reviewer #2: Yes

5. Review Comments to the Author

Reviewer #1: The submission tackles a very interesting question but fails in several ways. (a) The choice of spatial unit is not sufficiently justified. (b) Scaling is not about size but about how interactions are affected by scale. Thus one would not expect similar scaling behavior across spatial units which might represent different interactions. (c) The methodological choice of a linear spline at thre population mean is not justified.

Reviewer #2: 1. The manuscript is technically sound. The content is informative, new and interesting to public health institutions and also individuals seen the huge problematics of opioide misuse in the US.

2. I am no statistics expert, so I can not really evaluate the statistics topic.

3. Neverthesless from my limited understanding of statistics the data seem to be complete.

4. The manuscript is understandable and written in standard English (I found only some small typos).

Due to my limited understanding of statistics I suggest a minor revision by a person with more statistical knowledge.

6. PLOS authors have the option to publish the peer review history of their article (what does this mean?). If published, this will include your full peer review and any attached files.

Reviewer #1: No

Reviewer #2: No

---

## [Author Response · Author response to Decision Letter 0]

3 Nov 2020

Response to reviewers

We thank the editor and the reviewers for their comments. We appreciate your time and are confident that this version of our manuscript is significantly superior to the original version submitted. Please see below a point-by-point response to the comments.

Editor:

Comment: It is an important topic, and the results would deserve to have a nice publication.

However, the paper misses in my point of view the theoretical hypotheses that would strengthen the interpretation. In particular, it does not contextualize the relation between medical treatments / deseases / medical infrastructure. It is argued that there are no studies on scaling and health that is not true: see for example L. E. C. Rocha, A. E. Thorson, R. Lambiotte (2015). The non-linear health consequences of living in larger cities, Journal of Urban Health 92 (5) and some others since that publication....

so please highly strength the state of the art and the reflexion (fllowing also reviewer 1)

Response: Thank you for your comments. We have now added information to clarify the mechanisms underlying the scaling phenomenon of various outcomes. In the case of health outcomes, we also clarified the relationships between access to medical resources and the scaling behaviors of health outcomes. We also included other studies examining the scaling of health outcomes, including Rocha et al. (pages 4-5) See version with track changes

Reviewer 1: 

Comment: (a)The submission tackles a very interesting question but fails in several ways. (a) The choice of spatial unit is not sufficiently justified. 

Response: Thank you for your comments. We have now added a justification for the use of commuting zones (page 5, paragraph 2, and page 12). See version with track changes.

Comment: (b) Scaling is not about size but about how interactions are affected by scale. Thus one would not expect similar scaling behavior across spatial units which might represent different interactions. 

Response: Thank you for pointing this out. We have now added information to clarify the mechanisms underlying the scaling phenomenon of various outcomes (pages 3-5). In the case of health outcomes, we also clarified the relationships between access to medical resources and scaling behaviors of some outcomes. We also included other studies examining the scaling of health outcomes. (pages 4-5). We agree that the use of different spatial units can affect the scaling behavior. We have indicated that in the limitation section (page 12). To test whether our choice of spatial unit had an influence on the scaling behavior, we replicated the analysis using a different spatial definition– the Core-based Statistical Areas (CBSA). (S1 Fig 3). This analysis showed consistent result with those obtained from commuting zones.

Comment: (c) The methodological choice of a linear spline at the population mean is not justified.

Response: We have now added the justification for this choice in the method section (page 6-7)

Reviewer 2:

Comment: The manuscript is technically sound. The content is informative, new and interesting to public health institutions and also individuals seen the huge problematics of opioide misuse in the US.

2. I am no statistics expert, so I can not really evaluate the statistics topic.

3. Neverthesless from my limited understanding of statistics the data seem to be complete.

4. The manuscript is understandable and written in standard English (I found only some small typos).

Response: Thank you for your comments. We have carefully reviewed the manuscript and fixed these typos.

---

## [Decision Letter · Decision Letter 1]

14 Jan 2021

PONE-D-20-08832R1

Urban scaling of opioid analgesic sales in the United States

PLOS ONE

Dear Dr. Mullachery,

Thank you for submitting your manuscript to PLOS ONE. After careful consideration, we feel that it has merit but does not fully meet PLOS ONE’s publication criteria as it currently stands. Therefore, we invite you to submit a revised version of the manuscript that addresses the points raised during the review process.

Thanks for the improvement of the paper. Please finalize this improvement following the recommendations of the Reviewer 3.

We look forward to receiving your revised manuscript.

Kind regards,

Celine Rozenblat

Academic Editor

PLOS ONE

Additional Editor Comments (if provided):

Thanks for the improvement of the paper. Please finalize this improvement following the recommendations of the Reviewer 3.

Reviewers' comments:

Reviewer's Responses to Questions

**Comments to the Author**

1. If the authors have adequately addressed your comments raised in a previous round of review and you feel that this manuscript is now acceptable for publication, you may indicate that here to bypass the “Comments to the Author” section, enter your conflict of interest statement in the “Confidential to Editor” section, and submit your "Accept" recommendation.

Reviewer #2: All comments have been addressed

Reviewer #3: (No Response)

2. Is the manuscript technically sound, and do the data support the conclusions?

Reviewer #2: Yes

Reviewer #3: Yes

3. Has the statistical analysis been performed appropriately and rigorously? 

Reviewer #2: I Don't Know

Reviewer #3: N/A

4. Have the authors made all data underlying the findings in their manuscript fully available?

Reviewer #2: Yes

Reviewer #3: (No Response)

5. Is the manuscript presented in an intelligible fashion and written in standard English?

Reviewer #2: Yes

Reviewer #3: Yes

6. Review Comments to the Author

Reviewer #2: It seems that there has been a substantial improvement in content and, as mentioned by reviewer 1,

clearing of other available contextual literature and research. It also seems that all data are fully available and accesible on the gitHub platforme. In the abstract there is still a small typo (line 14: superlinear (not superliner). Soures / literature are added and also explained. The structure of the document is now much clearer. (All this under the precondition that I am not an expert in statistics neither in opioid/medical-social topics).

Reviewer #3: Report on “Urban scaling of opioid analgesic sales in the United States”

The authors analyze the prescription of opioid pills in commuting zones (CZ) in the USA. Analyzing the entire set of CZ they find super-linear scaling, ie in large cities more pills are prescribed. However, the authors find that the residuals exhibit a systematic deviation in a U-shape from which the authors infer two different scaling regimes. Separating the CZ into two groupd according to the median, a super-linear (below the median) and a super-linear (above the median) regime are found. The authors hypothesize reasons for this different scaling regimes.

Overall, this is a nice little paper. It is mostly well written and relevant to the urban scaling community and probably also for the opioid-crisis community. Accordingly, I recomment publication.

The only (non-mandatory) thing that the authors could consider is a better statistical treatment. They could automatically find a best division value for the two regimes (instead of the median), ie an optimization. In addition, colleagues with statistics background would appreciate seeing some test statistics that the model with two regimes fits better than the model with only one beta. In this context Akaike Information Criterion might be useful.

Specific comments:

- which location is use, address of patient, doctor, or pharmacy?

- “Scaling is the response of complex systems, such as cities, to changes in their size.” might be misleading. In most cases urban scaling is studied cross-sectionally (fixed year), but “changes” suggests change over time

- beginning of page.11: “For example” is just repeating what is already said in the previous sentence

- “disporportionally high number of social conections in large cities leads to an exponential increase in various outcomes such as economic productivity and number of patents”: the increase is probably not exponential

- why 607 CZ? Is this the total number? If not, how have they been chosen, why have others been omitted?

- “After visually exploring initial results, we detected a strong non-linear pattern” In log-log representation, I assume

- how do the authors deal with zero-values? Ie are there any CZ with no pills? If they, then the log-value cannot be shown

- why are the Figures in the SI?

- “we added a linear fit with a linear spline at the population median” spline is not visible, is the linear regressions are discontinuous

- Tab.1: it is not clear how adjustment and stratification has been done

- somehow the authors describe more Figures than were in my pdf

- to me the probable better availability of illegal drugs in large cities (the third reason mentioned by the authors) sounds most plausible for the sub-linear scaling of CZ above the median

- how do the years 2006-2014 go into the analysis? Is it that the values simply represent the total pills prescribed in this period?

7. PLOS authors have the option to publish the peer review history of their article (what does this mean?). If published, this will include your full peer review and any attached files.

Reviewer #2: No

Reviewer #3: No

---

## [Author Response · Author response to Decision Letter 1]

1 Feb 2021

Response to reviewers.

Thank you for your comments. Below we provide a point-by-point response to the comments.

Reviewer #2: It seems that there has been a substantial improvement in content and, as mentioned by reviewer 1, clearing of other available contextual literature and research. It also seems that all data are fully available and accessible on the gitHub platforme. In the abstract there is still a small typo (line 14: superlinear (not superliner). Soures / literature are added and also explained. The structure of the document is now much clearer. (All this under the precondition that I am not an expert in statistics neither in opioid/medical-social topics).

R: Thank you so much for your careful review of our manuscript. We have fixed the typo indicated.

Reviewer #3: Report on “Urban scaling of opioid analgesic sales in the United States”

The authors analyze the prescription of opioid pills in commuting zones (CZ) in the USA. Analyzing the entire set of CZ they find super-linear scaling, ie in large cities more pills are prescribed. However, the authors find that the residuals exhibit a systematic deviation in a U-shape from which the authors infer two different scaling regimes. Separating the CZ into two groups according to the median, a super-linear (below the median) and a super-linear (above the median) regime are found. The authors hypothesize reasons for this different scaling regimes.

Overall, this is a nice little paper. It is mostly well written and relevant to the urban scaling community and probably also for the opioid-crisis community. Accordingly, I recommend publication.

The only (non-mandatory) thing that the authors could consider is a better statistical treatment. They could automatically find a best division value for the two regimes (instead of the

R: Thank you so much for your careful review. We agree with the reviewer and have re-run the analysis with the spline using a data-driven approach. Specifically, we have now explored for the best-fitting spline knot, selected as the knot that maximizes model fit. While results are not substantially different from our original findings, we have now reported this as the main analysis. We have also provided a measure of fit (AIC) between the no-spline model and the model with a spline, finding a much better fit in the spline model. 

Specific comments:

- which location is use, address of patient, doctor, or pharmacy?

R: We used the location of the pharmacy or clinic. We have added sentences to make that clean in the text on pages 5 (last paragraph) and 6 (paragraph 2). See version with track changes.

- “Scaling is the response of complex systems, such as cities, to changes in their size.” might be misleading. In most cases urban scaling is studied cross-sectionally (fixed year), but “changes” suggests change over time.

R: The have changed to “variations” in size.

- beginning of page.11: “For example” is just repeating what is already said in the previous sentence.

R: We have now removed this sentence as it was, as indicated by the reviewer, completely redundant. See paragraph 3 of the introduction.

- “disporportionally high number of social connections in large cities leads to an exponential increase in various outcomes such as economic productivity and number of patents”: the increase is probably not exponential.

R: We meant to say: ”high number of social connections in large cities leads to a disproportional increase in various outcomes…” Thank you for identifying this issue. We have now fixed it. See track changes in paragraph 3 of the introduction.

- why 607 CZ? Is this the total number? If not, how have they been chosen, why have others been omitted?

R: 607 are all CZs in the continental US. We have clarified that and explained the rationale in the first paragraph of the methods.

- “After visually exploring initial results, we detected a strong non-linear pattern” In log-log representation, I assume.

R: Yes. We have clarified that in the fourth paragraph of the method section.

- how do the authors deal with zero-values? Ie are there any CZ with no pills? If they, then the log-value cannot be shown.

R: All CZs had at least one pill so there were no CZs with zero values. The number of CZs (607) in the final model can be seen on the tables. We have also added that information in the title of Figure 1.

- why are the Figures in the SI?

R: SI (Supplementary Information) contain the figures and tables that resulted from the sensitivity or exploratory analysis that were not part of the main analysis. Regarding the main paper figures, it may be that PLOS ONE is grouping them after the SI in the review PDF.

- “we added a linear fit with a linear spline at the population median” spline is not visible, is the linear regressions are discontinuous.

R: The linear regressions are discontinuous because they represent two different slopes: one for the CZs above the median and the other for the CZs below the median. However, to make this clearer, we have added a flag on the main figure indicating where the knot position is.

- Tab.1: it is not clear how adjustment and stratification has been done.

R: We have added information about the adjustment and region-specific analysis on page 7 (paragraph 2). See version with track changes. For the adjustment we used dummy variables for regions. For the stratified analysis, we estimated coefficients for each region separately. We have also added a footnote to table 1.

- somehow the authors describe more Figures than were in my pdf.

R: We are sorry about this. This issue may emerge from some figures that are in the supplementary information. We have now corrected this. We have also re-ordered the description of the Figures and Tables in the result section to prioritize results from the main analysis followed by results from the sensitivity analysis (which were included in the SI).

- how do the years 2006-2014 go into the analysis? Is it that the values simply represent the total pills prescribed in this period?

R: We summed all pills for the period for each CZ. We have now added this information in the third paragraph of the methods section.

---

## [Decision Letter · Decision Letter 2]

8 Apr 2021

PONE-D-20-08832R2

Urban scaling of opioid analgesic sales in the United States

PLOS ONE

Dear Dr. Mullachery,

Thank you for submitting your manuscript to PLOS ONE. After careful consideration, we feel that it has merit but does not fully meet PLOS ONE’s publication criteria as it currently stands. Therefore, we invite you to submit a revised version of the manuscript that addresses the points raised during the review process.  One reviewer noted a few minor suggestions which could further strengthen the manuscript.  

We look forward to receiving your revised manuscript.

Kind regards,

Nickolas D. Zaller

Academic Editor

PLOS ONE

Journal Requirements:

Reviewers' comments:

Reviewer's Responses to Questions

**Comments to the Author**

1. If the authors have adequately addressed your comments raised in a previous round of review and you feel that this manuscript is now acceptable for publication, you may indicate that here to bypass the “Comments to the Author” section, enter your conflict of interest statement in the “Confidential to Editor” section, and submit your "Accept" recommendation.

Reviewer #2: All comments have been addressed

2. Is the manuscript technically sound, and do the data support the conclusions?

Reviewer #2: Yes

3. Has the statistical analysis been performed appropriately and rigorously? 

Reviewer #2: I Don't Know

4. Have the authors made all data underlying the findings in their manuscript fully available?

Reviewer #2: Yes

5. Is the manuscript presented in an intelligible fashion and written in standard English?

Reviewer #2: Yes

6. Review Comments to the Author

Reviewer #2: It is visible see that you worked hard on the last version and answered to all comments of the reviewers. It seems to me (as already mentioned, I am not a statistics expert) that you did a lot of new research.

I found the discussion and explanations with the reviewer very interesting and helpful.

Maybe in general it is helpful to explain results in this way. It depends also on the target group how you formulate your text. Myself as a scientist with a backgroud in Geology/Paleontogy and Systems Scientist I am always working on making results easily understandable (also for "oursiders").

Therefore I very much appreciate your documentation of the discussion and review process.

Finally two small typos again in the downloaded Fig 2 & 3: I suppose the legend should be named "coefficient" (not "coefificient").

7. PLOS authors have the option to publish the peer review history of their article (what does this mean?). If published, this will include your full peer review and any attached files.

Reviewer #2: No

---

## [Author Response · Author response to Decision Letter 2]

14 Apr 2021

Response to reviewers

Thank you for your comments. Below we provide a point-by-point response to the comments.

Reviewer #2: It is visible see that you worked hard on the last version and answered to all comments of the reviewers. It seems to me (as already mentioned, I am not a statistics expert) that you did a lot of new research.

I found the discussion and explanations with the reviewer very interesting and helpful.

Maybe in general it is helpful to explain results in this way. It depends also on the target group how you formulate your text. Myself as a scientist with a backgroud in Geology/Paleontogy and Systems Scientist I am always working on making results easily understandable (also for "oursiders").

Therefore I very much appreciate your documentation of the discussion and review process.

Finally two small typos again in the downloaded Fig 2 & 3: I suppose the legend should be named "coefficient" (not "coefificient").

R: Thank you for your comments. We appreciate the time and effort put in this review. We genuinely believe that this has made our paper much stronger.

We have now fixed the typos in Fig 1 and supplemental Figs 2 and 3. Thank you.

---

## [Decision Letter · Decision Letter 3]

27 Jul 2021

PONE-D-20-08832R3

Urban scaling of opioid analgesic sales in the United States

PLOS ONE

Dear Dr. Mullachery,

Thank you for submitting your manuscript to PLOS ONE and for your patience during the review process. I am terribly sorry for the delay, it has been quite challenging securing reviewers during the current pandemic.  I have tried to do my due diligence in soliciting reviews for your revised manuscript after at least one prior reviewer was no longer available.  As you can see from the reviewer comments, there are still some points that need to be addressed in the revised manuscript.  I believe that these comments are addressable and therefore, I invite you to submit a revised manuscript. 

We look forward to receiving your revised manuscript.

Kind regards,

Nickolas D. Zaller

Academic Editor

PLOS ONE

Reviewers' comments:

Reviewer's Responses to Questions

**Comments to the Author**

1. If the authors have adequately addressed your comments raised in a previous round of review and you feel that this manuscript is now acceptable for publication, you may indicate that here to bypass the “Comments to the Author” section, enter your conflict of interest statement in the “Confidential to Editor” section, and submit your "Accept" recommendation.

Reviewer #4: (No Response)

Reviewer #5: All comments have been addressed

2. Is the manuscript technically sound, and do the data support the conclusions?

Reviewer #4: No

Reviewer #5: Yes

3. Has the statistical analysis been performed appropriately and rigorously? 

Reviewer #4: No

Reviewer #5: Yes

4. Have the authors made all data underlying the findings in their manuscript fully available?

Reviewer #4: Yes

Reviewer #5: Yes

5. Is the manuscript presented in an intelligible fashion and written in standard English?

Reviewer #4: Yes

Reviewer #5: Yes

6. Review Comments to the Author

Reviewer #4: 1. It would help to better justify the research approach. All prescription opioid counts from 2006 to 2014 were aggregated for each county over a period of increasing prescriptions. Why isn’t the outcome of interest the growth of prescriptions over the time period rather than the total amount? (It seems like from a policy perspective, you would want to know where prescriptions increased substantially given the introduction section.) Do any of the CZs grow substantially over the time period? If so, then this seems like information you would want to examine (if not, it justifies the averaging of population size and should be stated). That is, do fast growing populations correlate with fast growing prescriptions? Is this a confounder to worry about? What I’m getting at is the paper reads as if a couple variables were chosen without much thought to see if they were or weren’t correlated. Why is the chosen correlation the one to focus on rather than a different one? Explaining this would help justify the statistical approach.

2. It’s not clear why Alaska and Hawaii were excluded. Are they weird outliers for some reason? At a minimum the authors should explain more clearly why they should be excluded and test whether the estimates change in any meaningful way if Alaska and Hawaii are included. If there is no real difference, this should be stated. If the estimates do change in meaningful ways, then omitting them really needs a stronger justification.

3. The U-shaped pattern in Fig S1 seems largely driven by the three outlier residuals in the lower left of the figure. Indeed, when omitted in Table S1, the coefficient for beta1 falls a large amount. Although the “findings are robust” to this, meaning the signs of the coefficients and statistical significance presumably don’t change, the magnitude of the coefficient changes and needs to be discussed. Does the U-shape disappear? Does the optimal spline change? Are the outliers located in a particular region, perhaps explaining some of the stratification results?

4. In Table 1, what does “adjusted for region” mean? Does it mean a dummy variable was included for each region in the estimates? Or something else? This can be clarified in the text or Table 1 notes.

5. There is no interpretation of the coefficients. What does a beta1 of 1.42 vs 1.17 mean from a practical perspective? Both are superlinear, but does the difference of magnitude mean anything? How should they be interpreted? Why should we care about these specific numbers? Does a beta1 of 1.01 and a beta2 of 0.99 have any real meaning? If not, what level of these coefficients should matter? Or does it not matter and only the superlinear or not question is what matters (if so why?)?

6. If it is really a U-shaped relationship, why not use a quadratic estimation equation (add a population squared term)? Does it fit the data better? If so, how would you interpret the results?

7. The discussion needs work. The first sentence of the second paragraph is “Our results highlight the importance of exploring nonlinear scaling”, but nowhere is the “importance” explained. That it is detected in the estimates does not mean it is “important”. More care needs to be taken in explaining the importance of the results if this is the goal. (PLOS ONE does not judge things based on importance, so the highlighting of the importance could be removed if the author does not want to justify the importance of the results.)

8. Discussion says “One potential explanation [that below a threshold, higher populations are more strongly correlated with more opioid prescriptions, and above it, less] is that increases in potential matches beyond a certain threshold no longer translate into higher rates of successful matches.” But isn’t it that beyond some threshold, the higher rates of successful matches slows down, not that it disappears? Or am I missing something? It’s not like there is a sharp break at a threshold, but a (very) gradual flattening of the curve.

9. Several potential explanations are given in the discussion, but it would be helpful if it was explained why these (or other confounders) were not included in the analysis. Can you control for number of pharmacies/prescribers or not? Levels of prescription opioid vs heroin deaths in a CZ? It seems there are a potential long list of confounders that the limitations should highlight (poverty rates, unemployment, demographic profile, number of physicians, income level, existence of PDMP programs, opioid treatment availability, etc.). In addition, since PLOS ONE requires "appropriate controls", the lack of any controls needs justification.

Small typo: “42th” is used instead of 42nd.

Reviewer #5: The authors did a great job addressing previous reviewers comments. The revisions helped to clarify the methods and results.

On page 5, line 6, "maybe" should be "may be"

On page 10, line 7, "Last" should be "Lastly"

On page 11, line 3, insert "be" between "could" and "the"

On page 12, line 12, insert a reference for the statement about patients filling prescriptions close to home or work.

7. PLOS authors have the option to publish the peer review history of their article (what does this mean?). If published, this will include your full peer review and any attached files.

Reviewer #4: No

Reviewer #5: No

---

## [Author Response · Author response to Decision Letter 3]

18 Aug 2021

Response to reviewers

Thank you for your comments. Below we provide a point-by-point response.

Reviewer #4: 1. It would help to better justify the research approach. All prescription opioid counts from 2006 to 2014 were aggregated for each county over a period of increasing prescriptions. Why isn’t the outcome of interest the growth of prescriptions over the time period rather than the total amount? (It seems like from a policy perspective, you would want to know where prescriptions increased substantially given the introduction section.) Do any of the CZs grow substantially over the time period? If so, then this seems like information you would want to examine (if not, it justifies the averaging of population size and should be stated). That is, do fast growing populations correlate with fast growing prescriptions? Is this a confounder to worry about? What I’m getting at is the paper reads as if a couple variables were chosen without much thought to see if they were or weren’t correlated. Why is the chosen correlation the one to focus on rather than a different one? Explaining this would help justify the statistical approach.

Response:

Thank you for your comments. The growth in opioid prescription has been described elsewhere in the vast literature about the US opioid epidemic (see for example Jalal Science 2018, describing a long-term exponential growth in opioid deaths). Instead, the aim of our paper was to understand how opioid prescriptions scale with city size by leveraging the urban scaling framework. From the perspective of the population size, population did not change considerably in the period studied (median relative change from 2006 to 2014=+2.3% [IQR: -0.2% to 5.4%]), playing into our decision to use the pooled period rather than the year-by-year trend. Population growth can also be an important driver of health outcomes. This issue is being examined in other papers we are currently working on, but the examination of this phenomenon extends beyond the aim of this paper.

The variable population size was chosen after much thought considering the urban scaling framework. Scaling is the response of complex systems, such as cities, to variation in their size. The application of this framework has previously shown that a set of scaling relations can be used to predict several features of cities. Specifically, for the outcome opioid pills, one possible mechanism explaining the relationship between population size and opioid pills is that a disproportionally large number of social connections in relatively larger CZs creates an environment that facilitates a larger than expected number of successful matches between prescribers and patients, which in turn may lead to superlinear scaling of opioid analgesic pills. We explain these relationships in detail in the introduction and discussion of the paper. We have now made several edits to the introduction to make the aim and the theoretical framework more explicit.

2. It’s not clear why Alaska and Hawaii were excluded. Are they weird outliers for some reason? At a minimum the authors should explain more clearly why they should be excluded and test whether the estimates change in any meaningful way if Alaska and Hawaii are included. If there is no real difference, this should be stated. If the estimates do change in meaningful ways, then omitting them really needs a stronger justification.

Response:

We excluded Alaska and Hawaii because the commuting patterns in these states are expected to be different from those in the continental US since they are not connected by land to other states. Commuting patterns are key in the definition of Commuting Zones (CZs), one of the geographic delimitations we used in this paper. We have the justification in the text of the article. “We excluded CZs that include counties in non-contiguous states (Alaska and Hawaii) because they may not be a good representation of these networks that, in the continental US, often cross state lines.”

3. The U-shaped pattern in Fig S1 seems largely driven by the three outlier residuals in the lower left of the figure. Indeed, when omitted in Table S1, the coefficient for beta1 falls a large amount. Although the “findings are robust” to this, meaning the signs of the coefficients and statistical significance presumably don’t change, the magnitude of the coefficient changes and needs to be discussed. Does the U-shape disappear? Does the optimal spline change? Are the outliers located in a particular region, perhaps explaining some of the stratification results?

Response:

We thank the reviewer for this important insight. These three outliers are three small commuting zones (<10,000 pop) in New Mexico, South Dakota, and Montana. We have modified Fig S1 to reflect how the u-shape changes after removing these outliers. The pattern above for larger cities remains unchanged, but the pattern for smaller cities is now flat. This indicates that while the model without a spline is a good fit to smaller commuting zones, it still fails to reproduce patterns of opioid prescribing in larger cities. We have now reflected this in the text and in Figure S1. We have also explored whether the spline knot changes, and indeed it does change from 81,000 to 151,000 (very close to the median of 154,000). We have now added an extra robustness check by showing the scaling coefficients without the outliers and with the new threshold. As with the previous sensitivity analysis, the direction (and significance) of coefficients remains unchanged, although there’s a weaker superlinearity in smaller commuting zones (going from 1.36 in the main analysis to 1.24 after excluding outliers and changing the knot location). 

4. In Table 1, what does “adjusted for region” mean? Does it mean a dummy variable was included for each region in the estimates? Or something else? This can be clarified in the text or Table 1 notes.

Response:

We have added a footnote in Table 1 indicating that we included dummy variables for region (or stratified, in the case of stratified models):

“b Models adjusted for region included dummy variables for each region.

c Stratified models included the CZs for each region separately.”

5. There is no interpretation of the coefficients. What does a beta1 of 1.42 vs 1.17 mean from a practical perspective? Both are superlinear, but does the difference of magnitude mean anything? How should they be interpreted? Why should we care about these specific numbers? Does a beta1 of 1.01 and a beta2 of 0.99 have any real meaning? If not, what level of these coefficients should matter? Or does it not matter and only the superlinear or not question is what matters (if so why?)?

Response:

We have now added the interpretation to the results. See results: 

“The scaling coefficient for opioid analgesic pills in all 607 US Commuting Zones from 2006 to 2014 was 1.08 (95% CI 1.05-1.11), corresponding to superlinear scaling. These results show that the number of analgesic opioid pills was disproportionately higher in large (vs. small) CZs (S2 Fig.). Specifically, a CZ with 1% larger population had 1.08% greater pill count. However, we found that the model introducing a spline had a better fit than the model without a spline (AIC=127.8 in the model with a spline vs AIC=191.3 in the model without a spline), indicating a non-linear scaling behavior. Fig. 1 shows that CZs with population below the knot (population of 82,363) scale superlinearly (β=1.36, 95%CI 1.23 to 1.50), and CZs with population above the knot scale sublinearly (β=0.92, 95%CI 0.88 to 0.95). This means that for CZs below the knot, a 1% larger CZ had a 1.36% higher pill count, while for CZs above the knot, a 1% larger CZ had a 0.92% higher pill count.” 

A superlinear scaling coefficient of 1.08 means that a CZ with a 1% larger population had a 1.08% greater pill count, meaning that that the pill count was disproportionally larger even after accounting for the fact that the CZ has greater population. A larger coefficient (or smaller in the case of sublinear scaling) means stronger scaling which is important to characterize the behavior of cities regarding specific outcomes.

6. If it is really a U-shaped relationship, why not use a quadratic estimation equation (add a population squared term)? Does it fit the data better? If so, how would you interpret the results?

We agree with the author that a quadratic polynomial for population (so log(pills)=b0+b1*logpopulation+b2*logpopulation^2) would also probably fit the data best. However, interpreting such coefficients is challenging, as they would no longer provide an estimate of the scaling behavior of opioid pill sales. Our approach is a version of the “power law with cut-off” approach described in Clauset et al. (SIAM 2009). We have now referenced this on the methods section.

7. The discussion needs work. The first sentence of the second paragraph is “Our results highlight the importance of exploring nonlinear scaling”, but nowhere is the “importance” explained. That it is detected in the estimates does not mean it is “important”. More care needs to be taken in explaining the importance of the results if this is the goal. (PLOS ONE does not judge things based on importance, so the highlighting of the importance could be removed if the author does not want to justify the importance of the results.)

Response:

We adjusted the discussion to address this issue by not mentioning the importance of exploring nonlinear scaling. This paragraphs now reads: “Our results show the nonlinear scaling behaviour of opioid pills, a pattern that has been found in other outcomes”. The goal of the paper was to explore the distribution of pills using the urban scaling framework. We believe this can be important to understand larger patterns in cities. 

8. Discussion says “One potential explanation [that below a threshold, higher populations are more strongly correlated with more opioid prescriptions, and above it, less] is that increases in potential matches beyond a certain threshold no longer translate into higher rates of successful matches.” But isn’t it that beyond some threshold, the higher rates of successful matches slows down, not that it disappears? Or am I missing something? It’s not like there is a sharp break at a threshold, but a (very) gradual flattening of the curve.

Response:

Thank you for identifying this issue. We have adjusted the discussion. It now says: “One potential explanation is that the rate of successful matches decreases beyond a certain threshold”.

9. Several potential explanations are given in the discussion, but it would be helpful if it was explained why these (or other confounders) were not included in the analysis. Can you control for number of pharmacies/prescribers or not? Levels of prescription opioid vs heroin deaths in a CZ? It seems there are a potential long list of confounders that the limitations should highlight (poverty rates, unemployment, demographic profile, number of physicians, income level, existence of PDMP programs, opioid treatment availability, etc.). In addition, since PLOS ONE requires "appropriate controls", the lack of any controls needs justification.

Response:

Variables such as number of pharmacies, number of physicians, and poverty are hypothesized to be part of the mechanism for why population size is associated with opioid pills. Adjusting for these variables would block the mechanisms that lead to the association. For example, a larger population may lead to a larger concentration of physicians which in turn leads to a larger number of prescriptions. Given that we want to examine the broader relationship between population and outcome and describe the world as is, we believe that these variables should not be adjusted for. Future studies looking to explain mechanisms behind these patterns can explore whether scaling behaviors are attenuated after controlling for these variables. We have added this justification in the method section: “We used the model described above following standard practice in the urban scaling literature and did not adjust for any variables that may be in the pathway between population size and the outcome.”. We have also indicated in the discussion that future studies may want to explore these mechanisms.

Small typo: “42th” is used instead of 42nd.

Response: Thank you. We have now fixed that.

Reviewer #5: The authors did a great job addressing previous reviewers comments. The revisions helped to clarify the methods and results.

On page 5, line 6, "maybe" should be "may be"

On page 10, line 7, "Last" should be "Lastly"

On page 11, line 3, insert "be" between "could" and "the"

On page 12, line 12, insert a reference for the statement about patients filling prescriptions close to home or work.

Response:

Thank you for your additional comments. We have now fixed the issues that you raised. We have re-phrased the sentence on page 12, line 12, to reflect a general statement about commuting zones (vs. a specific statement about where patients fill their prescriptions, for which we do not have data).

---

## [Decision Letter · Decision Letter 4]

30 Sep 2021

Urban scaling of opioid analgesic sales in the United States

PONE-D-20-08832R4

Dear Dr. Mullachery,

We’re pleased to inform you that your manuscript has been judged scientifically suitable for publication and will be formally accepted for publication once it meets all outstanding technical requirements.

Kind regards,

Nickolas D. Zaller

Academic Editor

PLOS ONE

Additional Editor Comments (optional):

Reviewers' comments:

Reviewer's Responses to Questions

**Comments to the Author**

1. If the authors have adequately addressed your comments raised in a previous round of review and you feel that this manuscript is now acceptable for publication, you may indicate that here to bypass the “Comments to the Author” section, enter your conflict of interest statement in the “Confidential to Editor” section, and submit your "Accept" recommendation.

Reviewer #4: All comments have been addressed

2. Is the manuscript technically sound, and do the data support the conclusions?

Reviewer #4: Yes

3. Has the statistical analysis been performed appropriately and rigorously? 

Reviewer #4: Yes

4. Have the authors made all data underlying the findings in their manuscript fully available?

Reviewer #4: Yes

5. Is the manuscript presented in an intelligible fashion and written in standard English?

Reviewer #4: Yes

6. Review Comments to the Author

Reviewer #4: The authors have addressed all of my comments. The authors have clarified their statistical approach and discussion.

7. PLOS authors have the option to publish the peer review history of their article (what does this mean?). If published, this will include your full peer review and any attached files.

Reviewer #4: No

---

## [Editor Report · Acceptance letter]

4 Oct 2021

PONE-D-20-08832R4 

Urban scaling of opioid analgesic sales in the United States 

Dear Dr. Mullachery:

I'm pleased to inform you that your manuscript has been deemed suitable for publication in PLOS ONE. Congratulations! Your manuscript is now with our production department. 

Kind regards, 

on behalf of

Dr. Nickolas D. Zaller 

Academic Editor

PLOS ONE